# Applying a Wavelet Transform Technique to Optimize General Fitting Models for SM Analysis: A Case Study in Downscaling over the Qinghai–Tibet Plateau

**Zixuan Hu** [1], **Linna Chai** [1,*] , **Wade T. Crow** [2], **Shaomin Liu** [1], **Zhongli Zhu** [1], **Ji Zhou** [3] , **Yuquan Qu** [4], **Jin Liu** [1], **Shiqi Yang** [1] **and Zheng Lu** [1]

1. State Key Laboratory of Earth Surface Processes and Resource Ecology, Faculty of Geographical Science, Beijing Normal University, Beijing 100875, China; 201921051177@mail.bnu.edu.cn (Z.H.); smliu@bnu.edu.cn (S.L.); zhuzl@bnu.edu.cn (Z.Z.); liuj@mail.bnu.edu.cn (J.L.); 201821051088@mail.bnu.edu.cn (S.Y.); legend.lz@mail.bnu.edu.cn (Z.L.)
2. Hydrology and Remote Sensing Laboratory, Agricultural Research Service, United States Department of Agriculture, Beltsville, MD 20705, USA; wade.crow@usda.gov
3. School of Resources and Environment, Center for Information Geoscience, University of Electronic Science and Technology of China, Chengdu 611731, China; jzhou233@uestc.edu.cn
4. Forschungszentrum Jülich, Institute of Bio- and Geosciences: Agrosphere (IBG-3), 52428 Jülich, Germany; y.qu@fz-juelich.de
* Correspondence: chai@bnu.edu.cn

**Abstract:** Soil moisture (SM) is an important land-surface parameter. Although microwave remote sensing is recognized as one of the most appropriate methods for retrieving SM, such retrievals often cannot meet the requirements of specific applications because of their coarse spatial resolution and spatiotemporal data gaps. A range of general models (GMs) for SM analysis topics (e.g., gap-filling, forecasting, and downscaling) have been introduced to address these shortcomings. This work presents a novel strategy (i.e., optimized wavelet-coupled fitting method (OWCM)) to enhance the fitting accuracy of GMs by introducing a wavelet transform (WT) technique. Four separate GMs are selected, i.e., elastic network regression, area-to-area regression kriging, random forest regression, and neural network regression. The fitting procedures are then tested within a downscaling analysis implemented between aggregated Global Land Surface Satellite products (i.e., LAI, FVC, albedo), Thermal and Reanalysis Integrating Medium-resolution Spatial-seamless LST, and Random Forest Soil Moisture (RFSM) datasets in both the WT space and the regular space. Then, eight fine-resolution SM datasets mapped from the trained GMs and OWCMs are analyzed using direct comparisons with in situ SM measurements and indirect intercomparisons between the aggregated OWCM-/GM-derived SM and RFSM. The results demonstrate that OWCM-derived SM products are generally closer to the in situ SM observations, and better capture in situ SM dynamics during the unfrozen season, compared to the corresponding GM-derived SM product, which shows fewer time changes and more stable trends. Moreover, OWCM-derived SM products represent a significant improvement over corresponding GM-derived SM products in terms of their ability to spatially and temporally match RFSM. Although spatial heterogeneity still substantially impacts the fitting accuracies of both GM and OWCM SM products, the improvements of OWCMs over GMs are significant. This improvement can likely be attributed to the fitting procedure of OWCMs implemented in the WT space, which better captures high- and low-frequency image features than in the regular space.

**Keywords:** soil moisture; downscaling; optimization; wavelet transform; Qinghai–Tibet Plateau

## 1. Introduction

Soil moisture (SM) is a fundamental hydrological variable. It not only shows great significance in the hydrological, bioecological, and biogeochemical cycles [1–3], but also plays an active role in hydrological processes such as precipitation, runoff, infiltration, and

evapotranspiration [4,5]. Operational SM products have been applied to many research fields, including global climate change [6], agricultural applications [7], drought and flood disaster monitoring [8,9], and weather forecasting [10,11].

Multiple satellite missions—e.g., the Soil Moisture Ocean Salinity (SMOS) and Soil Moisture Active and Passive (SMAP) missions—provide operational SM products. However, their application is often limited by their coarse spatial resolution [12,13] and spatiotemporal gaps in their data products [14,15]. These shortcomings are often mitigated via the production of downscaled and/or gap-filled SM products. For example, the 3 km SMAP SM product (L2_SM_SP) is based on downscaling the enhanced Level 2 SMAP brightness temperature product (L1C_TB_E) using 3 km co-polarized and cross-polarized backscatter measurements provided by Sentinel-1 radars [16]. By applying the semi-physical disaggregation based on physical and theoretical scale change (DisPATCh) method, a ground segment with a 1 km spatial resolution for the SMOS data—known as CATDS—has been developed over the global cover [17]. Likewise, the Climate Change Initiative (CCI) SM product minimizes temporal gaps by combining SM retrievals obtained from multiple active and passive sensors using weighted averaging [15].

In addition, a large number of empirical or semi-physical relationships have been developed to link coarse-resolution, satellite-based SM retrievals with other land-surface parameters to obtain finer-resolution and more-continuous SM data products. These fitted models take a wide variety of forms, including optical and thermal temperature/vegetation feature space regression [18–20], active and passive microwave data fusion [16,21–23], machine learning [24–26], deep learning [27,28], geostatistical methods [29–31], and data assimilation methods [32–34]. ElSaadani et al. investigated the applicability of a convolutional long short-term memory network (ConvLSTM) algorithm for predicting SM and filling observational gaps in south Louisiana in the United States [28]. Prasad et al. designed a new multivariate sequential predictive model that utilizes the ensemble empirical mode decomposition (EEMD) algorithm hybridized with extreme learning machines (ELMs) to forecast soil moisture (SM) over weekly horizons [35]. Jin et al. developed a geographically weighted area-to-area regression kriging (GWATARK) model for the upstream region of the Heihe River Basin [36]. Wei et al. used gradient boosting decision tree regression (GBDT) with SMAP L3_SM_P to produce a 1 km SM product over the Qinghai–Tibet Plateau (QTP) [37]. Moreover, Liu et al. compared six downscaling strategies, including artificial neural networks, Bayesian estimation, decision trees (CART), nearest-neighbor algorithms, random forests, and support-vector machines [38]. Qu et al. compared the performances of five SM fitting models (i.e., multiple statistical regression, DisPATCh, random forest, Gaussian process regression, and area-to-area regression Kriging) for SM downscaling [39]. Generally, these methods are characterized by inherent advantages, disadvantages, background theories, suitable operating environments, and other specific assumptions, which have been thoroughly analyzed in [40] and [41,42].

As discussed above, current SM analyses (e.g., gap-filling, downscaling, and forecasting) generally rely on existing SM fitting methods, with some based on comparisons between multiple methods. However, few studies have focused on general method optimization, which is directly related to the subsequent accuracy of the resulting SM products (i.e., products after gap-filling, downscaling, and/or forecasting). A possible issue in existing SM fitting methods is that it is often difficult to capture data characteristics in feature space. For example, SM products generally contain unique image signal characteristics at different image frequencies in the time and space domains. In SM analysis applications, image signal characteristics at different scales—caused by scale-invariance of spatial heterogeneity—also cannot be ignored [43,44]. Therefore, characteristics at a specific spatial resolution cannot be similarly extrapolated to other SM resolutions.

In response to these general challenges, we propose a plausible strategy (i.e., optimized wavelet-coupled fitting method, or OWCM) to optimize the general SM fitting methods (GM) via the application of the wavelet transform (WT) technique. WT is a powerful technique for separating the high- and low-frequency feature information of an image when

characterizing spatial features across multiple scales [44]. It has been applied for remote sensing data fusion [45,46]. Detailed information obtained by WT decomposition can potentially improve upon regression fitting accuracy. In this work, eight fine-resolution SM datasets are generated using four separate GMs—elastic network regression, area-to-area regression Kriging, random forest regression, and neural network regression—each fitted both with and without application of the WT technique. Next, the advantage associated with applying WT to the general fitting models is analyzed by comparing SM products that are OWCM-derived (i.e., fit in the WT transform space) versus GM-derived (i.e., fit in regular space without WT) fine-resolution ($0.01° \times 0.01°$, hereafter referred to as 'fine-RES') SM datasets. Our key objectives are therefore to (1) apply WT to develop new robust, optimized fitting approaches based on four general fitting methods; (2) test the spatiotemporal heterogeneity of each in situ node, to select ground SM observations with high representativeness of the true values; and (3) evaluate the performances of the OWCM-derived and GM-derived fine-RES SM datasets against selected ground observations.

## 2. Materials and Methods

### 2.1. Study Area and In Situ Network Measurements

The QTP is the highest plateau in the world, and contains a unique ecohydrological and geographical environment [47]. It covers approximately 250 million $km^2$, ranging from 26.5° to 40°N in latitude and 73° to 105° in longitude. As shown in Figure 1, most of the region is at a high elevation, and the average altitude is 4000 m above sea level [48]. The environmental characteristics of the QTP are alpine, arid, and anoxic, making the plateau's ecological environment extremely fragile and sensitive. The combination of its high elevation and unique atmospheric, water, and energy circulation creates a series of 'high-cold' vegetation types distributed over the QTP [49,50]. For nearly half a century, the QTP has had a trend of warming and humidification, and frequent hydrometeorological change has been observed in this region [51].

SM datasets measured in situ, which can be downloaded from https://data.tpdc.ac.cn (accessed on 21 June 2022), were collected from three study sites (Naqu, Ngari, and Maqu). They belong to the CTP-SMTMN (Soil Moisture and Temperature Monitoring Network on the central TP) [52] and the Tibet-Obs (Tibetan Plateau Observatory of Plateau-Scale Soil Moisture and Soil Temperature) [53,54] observatory systems. Naqu has a cold, semi-arid climate and land cover dominated by alpine meadows. The Ngari region is also characterized by alpine meadows, but with a cold, arid environment. The Maqu region has a cold, humid climate and dense vegetation cover. The primary characteristics of these networks are summarized in Table 1. More details can be found in the above references. In this study, ground measurements were used for two purposes: (1) examining the spatial representativeness of measurements collected by each in situ SM sensor within the three networks, and (2) evaluating the eight fine-RES SM datasets generated from the four GMs by combining and non-combining with the WT technique.

**Table 1.** Summary of the basic information of the three in situ networks.

| Networks | Ngari | Naqu | Maqu |
|---|---|---|---|
| Datasets | Tibet-Obs | CTP-SMTMN | Tibet-Obs |
| Location in QTP | West | Central | Northeast |
| Total nodes | 18 | 57 | 20 |
| Measured SM depth | 5 cm | 0–5 cm | 5 cm |
| Measured time interval | 15 min/day | 30 min/day | 15 min/day |
| Used time coverage | | April 2016–September 2016 | |

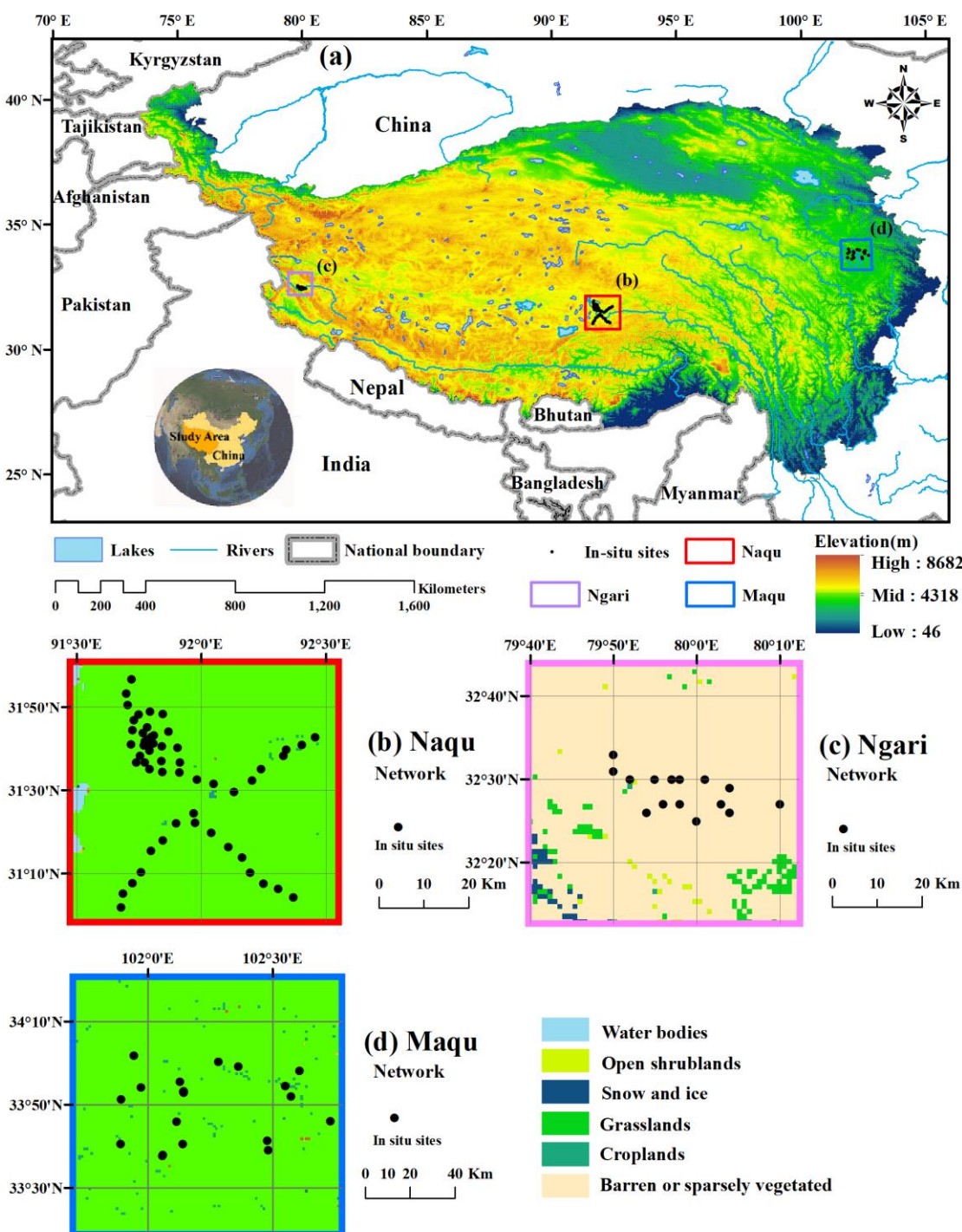

**Figure 1.** (**a**) The QTP study area and the (**b**) Naqu, (**c**) Ngari, and (**d**) Maqu in situ SM networks.

## 2.2. The 0.25° × 0.25° Original Soil Moisture Product

The 0.25° × 0.25° Random Forest SM product (RFSM, https://data.tpdc.ac.cn, accessed on 21 June 2022) is a reconstructed, long-term SM dataset with a satisfactory accuracy over the QTP [55,56]. Here, it was used as the coarse SM reference for constructing SM fitting models. The RFSM product was developed by adopting AMSR-E and AMSR2 microwave brightness temperature from five channels (H and V polarization of 10.7 GHz and 18.7 GHz, V polarization of 36.5 GHz) as well as auxiliary data (e.g., DEM, land cover, latitude, longitude) to train the random forest model on the first two years of the SMAP SM data product (2015 and 2016). Over the test period from May 2017 to May 2018,

RFSM preserved the traits of SMAP well (R = 0.95, RMSE = 0.03 m$^3$/m$^3$), based solely on AMSR-E and AMR2 inputs. Simultaneously, validation against in situ measurements showed that RFSM has relatively high temporal accuracy (R = 0.75, RMSE = 0.06 m$^3$/m$^3$, bias = −0.03 m$^3$/m$^3$). It is worth noting that higher uncertainties generally characterize remotely sensed SM products during the frozen season. With this limitation in mind, our analysis is based only on the unfrozen season from April to September 2016.

*2.3. The 0.01° × 0.01° Fine-RES Products*

Global Land Surface Satellite (GLASS, http://www.geodata.cn/thematicView/GLASS.html, accessed on 21 June 2022) datasets were used in this work. GLASS datasets include a series of high-precision, spatially continuous, long-term global products, i.e., leaf area index (LAI), surface broadband albedo (albedo), fractional vegetation cover (FVC), gross primary production (GPP), evapotranspiration (ET), etc. Here, the 1 km × 1 km GLASS FVC [57], 500 m × 500 m LAI [58], and albedo [59] products were adopted as independent variables to construct the fitting models. They were mosaicked and reprojected to cover the QTP using the MODIS Reprojection Tool. Next, nearest-neighbor interpolation was utilized to resample them to a spatial resolution of 0.01° × 0.01°. All pixels classified as water or ice/snow were removed.

Land-surface temperature (LST) is considered to be a dominant energy parameter describing surface water and heat conditions [29,60]. The 1 km Thermal and Reanalysis Integrating Medium-resolution Spatial-seamless LST–Tibetan Plateau (TRIMS LST-TP, https://data.tpdc.ac.cn, accessed on 21 June 2022) [61,62] was selected to overcome the challenges of thermal infrared remote sensing detection as a result of temporal/spatial gaps and misdetection due to cloud/topography causes [63,64]. TRIMS LST leverages both high spatial resolution from MODIS LST and all-weather capability from AMSR-E/AMSR2 passive microwave brightness temperature. Peng et al. demonstrated that LST is more sensitive to SM during daytime than nighttime, and has a stronger correlation with and sensitivity to SM than other factors [65]. Pablos et al. also concluded that downscaled SM results based on MODIS daytime LST are superior to results based on MODIS nighttime LST. As a result, the daytime TRIMS LST-TP product was applied here [66].

The above products, collected for training OWCMs and GMs during the unfrozen season from April to September 2016, were resampled and reprojected to fit the original RFSM SM product (0.25° × 0.25°) and target (0.01° × 0.01°) scales. Two additional factors derived from them—the temperature vegetation dryness index (TVDI) and soil evaporation efficiency (SEE)—were also adopted as independent variables for the fitting procedure. TVDI is a compound temperature–vegetation parameter derived from the soil moisture dry–wet edge contour in the two-dimensional feature space of vegetation index and LST [67–69]. SEE is a parameter obtained from the improved calculation method for soil moisture evaporation efficiency [29,70]. It is worth noting that the GLASS LAI from April 2014 to September 2016 was collected for in situ heterogeneity quantification, as shown in Section 3.2.1.

Given the known topographical influence on SM, topography complexity index (Tci) is a composite exponent computed from a digital elevation model (DEM). Here, DEM information, as shown in Figure 1a, was acquired from the USGS global 30-arc second elevation dataset (GTOPO-30) with a spatial resolution of 30″, developed by the United States Geological Survey (USGS, https://lta.cr.usgs.gov/, accessed on 21 June 2022). Tci was computed based on the following equation [56]:

$$Tci = \sqrt{\frac{1}{n}\sum_{i=1}^{n}\left(dem_i - \overline{dem}\right)^2}, \quad \overline{dem} = \frac{(dem_1 + \ldots + dem_n)}{n} \quad (1)$$

where $dem_i$, $\overline{dem}$, and $n$ are the $i$th, average value, and sample size of the 30″ DEMs within a 0.25° and 0.01° pixel, respectively. In addition, latitude, longitude, and DOY (day of year) information are also adopted as fitting parameters.

### 2.4. Auxiliary Data

The QTP land cover shown in Figure 1 is based on MODIS MCD12Q1 data and the International Geosphere-Biosphere Programme (IGBP) classification system. The QTP land cover data were used in the multifactorial statistic factor selection (MFS) process introduced in Appendix A.

The precipitation product (Pre) captures surface daily rainfall accumulations. These accumulations were extracted from the gridded daily scale dataset of CN05.1 (http://data.cma.cn/, accessed on 21 June 2022) [71] and used to validate the eight fine-RES SM datasets indirectly. As an augmentation of CN05 [72], CN05.1 is based on the interpolation of more node observations (~2400), and is characterized by a higher spatial resolution of $0.25° \times 0.25°$.

## 3. Methodology

### 3.1. Optimize General Fitting Methods by Wavelet Transform

Here, we introduce a method for optimizing general fitting models. Our key innovation is OWCM—that is, the application of a wavelet transform technique to four GMs to implement their fitting procedure in WT space. Elastic network regression, area-to-area regression kriging, random forest regression, and neural network regression were chosen to capture a range of GMs. As a test, we applied OWCMs to the specific SM problem of spatial downscaling. The selected four GMs were applied to both the regular and WT space to obtain four GM-derived and four OWCM-derived fine-RES SM datasets.

For a typical SM downscaling procedure implemented in the regular space, regression relationships are generally obtained between SM and attributes (e.g., time-varying FVC, LAI, albedo, LST, and time-constant DEM and LAT/LONG in this study) at a coarse resolution using a GM, and then applied to map between attributes and SM at a fine resolution. In contrast, OWCMs implement the downscaling procedures in the WT space. Specifically, (1) each of the coarse-resolution SM and the aggregated time-varying attributes were first decomposed to four WT components; (2) the GMs were then trained with each group of the same wavelet components obtained from SM and attributes at a coarse resolution; (3) the trained GMs were then utilized with the corresponding decomposed wavelet components of the attributes to map four SM wavelet components at a fine resolution; and (4) an inverse wavelet transformation was applied to the four fine-resolution wavelet components of SM to obtain the fine-RES SM image. It is worth noting that, before applying regression in the regular or WT space, multifactorial statistic factor selection (MFS) was implemented to determine the appropriate attributes. A detailed description of the MFS process is presented in Appendix A.

#### 3.1.1. Wavelet Transform

As a mathematical tool initially designed for signal processing, WT provides multiresolution/multiscale analysis functions and effectively extracts global and multiscale features of images within the spatial frequency domain [45].

The block diagram in Figure 2 explains the application of a wavelet transformation in detail. During the two-dimensional wavelet transform process, a digital image is decomposed into four images (hereafter called LL, HL, LH, and HH). Here, 'H' stands for high-pass filter, 'L' stands for low-pass filter, and 'HL' means a high-pass filter corresponding along the row direction of the input image and a low-pass filter following along the column direction. The same nomenclature applies to the LH, HH, and LL images. The spatial resolutions of these decomposed images are twice as coarse as the original one. For example, in this work, the spatial resolution of the original SM image is $0.25° \times 0.25°$, and the spatial resolution of its four decomposed images is $0.5° \times 0.5°$. Commonly applied WT techniques include the Haar, Daubechies, Coiflet, and Symmlet filters [73].

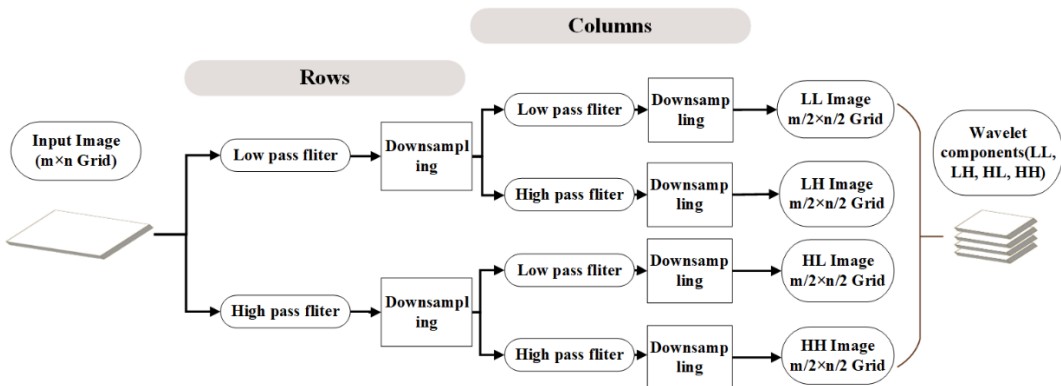

**Figure 2.** The block diagrams of the wavelet transformation in decomposing an $n \times m$ image into four decomposed images.

This study uses a 2D discrete Harr WT to decompose SM image signals. Following Figure 2, the Haar decomposition process can be expressed as shown in Equations (2) and (3) [74]. Taking an original $n \times m$ image $A(n, m)$ as an example, first, WT applies a convolution filter to matrix $A(n, m)$ along its row direction, and implements a downsampling process to obtain row decompositions of $A_L$ and $A_H$:

$$\begin{cases} A_{L(i,j)} = \frac{A_{(i,2j-1)} + A_{(i,2j)}}{2} & 1 \leq i \leq n,\ 1 \leq j \leq \frac{m}{2} \\ A_{H(i,j)} = \frac{A_{(i,2j-m-1)} - A_{L(i,j-\frac{m}{2})}}{2} & 1 \leq i \leq n,\ \frac{m}{2} < j \leq m \end{cases} \tag{2}$$

where $i$ and $j$ are the row and column number of $A$, respectively, and $A_L$ and $A_H$ represent the average and detail coefficients for image $A$, respectively.

Next, the Harr WT applies a convolution filter to $A_L$ and $A_H$ along the column direction:

$$\begin{cases} A_{LL(i,j)} = \frac{A_{L(2i-1,j)} + A_{L(2i,j)}}{2} & 1 \leq i \leq \frac{n}{2},\ 1 \leq j \leq \frac{m}{2} \\ A_{LH(i,j)} = \frac{A_{L(2i-n-1,j)} - A_{LL(i-\frac{n}{2},j)}}{2} & 1 \leq i \leq \frac{n}{2},\ \frac{m}{2} < j \leq m \\ A_{HL(i,j)} = \frac{A_{H(2i-1,j)} + A_{H(2i,j)}}{2} & \frac{n}{2} < i \leq n,\ 1 \leq j \leq \frac{m}{2} \\ A_{HH(i,j)} = \frac{A_{H(2i-n-1,j)} - A_{HL(i-\frac{n}{2},j)}}{2} & \frac{n}{2} < i \leq n,\ \frac{m}{2} < j \leq m \end{cases} \tag{3}$$

where $A_{LL}$ represents the low-frequency average characteristics, and looks most like the original image. Analogically, $A_{LH}$ represents the detail coefficients of $A_L$; $A_{HL}$ and $A_{HH}$ represent the average and detail coefficients of $A_H$, respectively. Note that fitting models can instead be applied to the WT space—e.g., the high-frequency details or low-frequency approximations obtained in Equation (3)—to retain high-frequency or low-frequency information contained in the original image. In addition, the four decomposition images generated in Equation (3) can be inversely reconstructed to regenerate image $A$ following the inverse process along the column and row directions.

### 3.1.2. General Fitting Methods

Elastic network regression (ER) [75] is a multivariate statistical regression model that uses two prior regularization terms from ridge regression and lasso regression. ER can eliminate the high collinearity between variables and reduce their dimensionality so as to obtain better fitting results than linear regression.

Area-to-area regression kriging (ATARK, hereafter referred to as ATA) [36] follows the principle that geographical attributes are often spatially autocorrelated [76]. The implementation of ATA includes two steps: (1) constructing the coarse-scale trend surface with the regression process and obtaining residuals by subtracting the regressed trend surface from the original geospatial field; and (2) applying area-to-area kriging interpolation

to reconstruct the residuals at a fine resolution and then superposing them onto the trend surface to obtain the final results.

The random forest model (RF), introduced by Breiman [77], is an ensemble learning technique commonly applied to classification and regression problems. The basic principle is integrating results taken from different decision trees through ensemble learning. Multiple decision trees are first built during training, and then the individual results of the trees are generated as the prediction. RF has many advantages, including insensitivity to multivariate collinearity and default hyperparameter values, with anti-noise ability, and strong robustness when applied to high-dimensional data [78,79].

To ensure that the training model was suitable for dealing with long spatiotemporal sequences and dynamic movement (e.g., SM), nonlinear autoregressive models with exogenous-input feedforward neural networks (NNs) were used for SM prediction. NNs have been shown to perform well on problems involving long-term dependencies [80]. In this approach, output signals from one layer are regressed on the previous layer's results with current and past values of the input signals by configuring a tapped delay line (TDL) [81,82].

### 3.2. Evaluation Strategy

The improvement of OWCMs over corresponding GMs was analyzed by direct comparisons of the GM-/OWCM-derived fine-resolution SM with in situ SM measurements, as well as indirect intercomparisons between the aggregated OWCM-/GM-derived SM and RFSM. To minimize point–cell-scale differences during direct validation, in situ measurements from grid cells with low heterogeneity were used. Moreover, the performances of GM- and OWCM-derived SM products were also evaluated in terms of their ability to spatially and temporally match original RFSM estimates.

### 3.2.1. Heterogeneity Quantification at the Fine-RES Scale

The weak spatial representativeness of point-scale in situ observations at satellite resolution scales represents a severe challenge for SM validation using ground measurements. Upscaling errors can be minimized by selecting ground stations that maximally represent a larger-scale region. Over the QTP, we identified in situ measurement sites with maximal representativeness by measuring the spatial heterogeneity of the corresponding spatial grid cell for better validation of the fine-RES SM.

To this end, coefficients of variation (CVs) and normalized information entropy (NIE) measures were utilized to analyze the degree of spatial heterogeneity present within a grid cell. CV is a normalized measure that characterizes data dispersion over a classical statistical index [83]. NIE, derived from information entropy [84], can clarify the relationship between probability and information redundancy.

Since vegetation and SM have the greatest impacts on the total grid heterogeneity, LAI and SM were adopted in the heterogeneity analysis. Following an approach previously described by Zhang et al. [85], the spatial heterogeneity score of a pixel was then described by their weighted CV and NIE values:

$$Score_{LAI} = 0.5 \times CV_{LAI} + 0.5 \times NIE_{LAI} \tag{4}$$

$$Score_{SM} = 0.5 \times CV_{SM} + 0.5 \times NIE_{SM} \tag{5}$$

where $CV_{LAI}$ and $NIE_{LAI}$ are the calculated CV and NIE of LAI, respectively, while $CV_{SM}$ and $NIE_{SM}$ are the corresponding values obtained from SM. The LAI and SM values used to obtain these statistics come from the corresponding GLASS LAI and ground-measured SM of the other four in situ measurement sites nearest to the evaluated one, considering days when in situ SM values were available.

Different weights were designed in calculating the total space heterogeneity score. The formula is as follows:

$$Score_{pixel} = 0.25 \times n(Score_{LAI}) + 0.75 \times n(Score_{SM}) \tag{6}$$

where $Score_{pixel}$ is the total space heterogeneity score of the fine-RES pixel where a single node is located, while $n(Score_{LAI})$ and $n(Score_{SM})$ represent the CV and NIE min–max scores normalized to 0–1, respectively. A smaller value of $Score_{pixel}$ indicates lower in situ spatial heterogeneity and, therefore, improved representativeness of the point-scale ground measurements within the pixel.

The heterogeneity of each pixel with the corresponding in situ SM measurement sites was determined during the frozen (from October to March) and unfrozen (from April to September) seasons. The daily scores were averaged to a seasonal value to measure temporal changes in heterogeneity through the seasons. Finally, the last few in situ measurement sites within each network (Naqu, Maqu, and Ngari) with the lowest-ranked pixel scores were selected. These in situ measurement sites were determined to have low spatiotemporal heterogeneity, revealing the high capacity to represent local ground 'true' values in point scale, and a good ability to evaluate the fine-RES SM.

### 3.2.2. Exploratory Data Analysis Method

Two time series may be strongly correlated within a specific period, but the correlation can be weakened during a sub-period. For example, the GM-/OWCM-derived multiyear SM sequences may be well correlated with RFSM due to their strong seasonality; however, the situations may differ within a short period—especially in a rain or irrigation season—because different fitting models, as well as different fitting strategies (e.g., the wavelet transformation), have different abilities in capturing the rainfall and dry-down events. Therefore, sampled correlation R sampled in different sub-periods may also be different. Here, the exploratory data analysis (EDA) method was used to analyze the temporal consistency between the OWCM-/GM-derived SM and RFSM within different sub-time-periods.

The EDA method was developed by Brunetti et al. [86], with the initial idea conceived for a methodology to study the correlation between two data series [87]. EDA allows for the representation of every variation of correlation between two series over time. In this work, EDA was utilized to examine the Pearson correlation coefficient variations between the fine-RES SM series of GMs/OWCMs and the original RFSM series during different time windows. The window length is defined as follows:

$$Length_{(i,j)} = \begin{cases} 2 \times (DOY_i - DOY_j) & DOY_{min} \leq DOY_j < DOY_i \leq DOY_{middle} \\ 2 \times (DOY_i - DOY_j) & DOY_{middle} < DOY_i \leq DOY_{max}, \ 2 \times DOY_i - DOY_{max} \leq DOY_j \end{cases}' \tag{7}$$

where $DOY_i$ and $DOY_j$ are the ordinals of the central and beginning days of the selected time series, respectively, while $DOY_{min}$, $DOY_{middle}$, and $DOY_{max}$ are the ordinals of the start, central, and end days of the entire time sequence, respectively.

### 3.3. Generalized Additive Model

To analyze the impacts of spatial heterogeneity on fitting accuracy, the contributions to the correlation coefficient R from $CV_{LST}$, $CV_{LAI}$, $CV_{TVDI}$, and $CV_{SEE}$ were explored based on a generalized additive model (GAM). Here, the correlation R substituted into GAM was calculated between aggregated GM-/OWCM-derived SM and RFSM in coarse $0.25° \times 0.25°$ spatial resolution throughout the study period. GAMs are useful when developing and evaluating the non-monotonic relationship between independent and dependent variables [88,89]. To better describe the contributions of the four CV variables in GAM, Tci and sample size were added as explanatory variables to the model. The expression of GAM was therefore constructed as follows:

$$Score_{pixel} = 0.25 \times n(Score_{LAI}) + 0.75 \times n(Score_{SM}) \tag{8}$$

where $c_0$ is an unknown coefficient; $x_i$ is the CV value of the $i$th continuous nonlinear explanatory variable; $f_{i,\ i=1,...,6}$ are nonparametric (unspecified) smooth functions between the correlation coefficient R and the nonlinear heterogeneity variables ($CV_{LAI}$, $CV_{LST}$, $CV_{TVDI}$, $CV_{SEE}$), Tci, and sample size; and $\varepsilon$ is the corresponding error. Explained deviance ($D_m$) is used to confirm the contribution of various explanatory variables to the total R, computed as follows:

$$D_m = 1 - D_r/D_n \tag{9}$$

where $D_r$ is the residual model deviance, and $D_n$ is the null model deviance. GAM is finally applied based on the six sequences described above (i.e., $CV_{LAI}$, $CV_{LST}$, $CV_{TVDI}$, $CV_{SEE}$, Tci, and sample size) using the R language package *mgcv* (https://www.rdocumentation.org/packages/mgcv/, accessed on 21 June 2022).

## 4. Results

### 4.1. The Spatiotemporal Heterogeneity Rankings of the $0.01° \times 0.01°$ Grids

The spatiotemporal heterogeneity of each $0.01° \times 0.01°$ grid cell containing an in situ measurement site within the Naqu, Maqu, and Ngari networks was quantified by the synthesis season scores of CV and NIE weighting summation, i.e., the $Score_{pixel}$ defined in Equation (6). This score was further utilized to determine the spatial representativeness of each in situ measurement site over the $0.01° \times 0.01°$ grid cell. The boxplot with scatters in Figure 3 indicates the heterogeneity distributions of all $0.01° \times 0.01°$ grid cells containing in situ measurement sites. Such grid cells were classified into three groups, with low (score $\leq 0.4$), moderate (0.4 < score $\leq 0.7$), and high (0.7 < score $\leq 1$) heterogeneity.

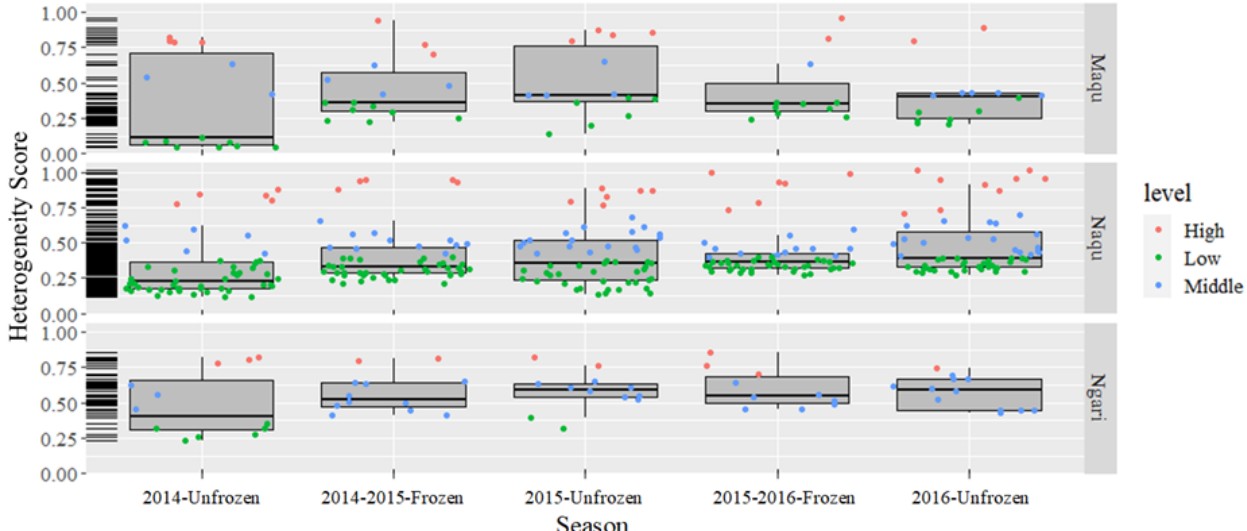

**Figure 3.** The heterogeneity distributions of all grid cells containing in situ measurement sites within the Naqu, Maqu, and Ngari networks. Green dots indicate low heterogeneity (score $\leq 0.4$), blue dots indicate moderate heterogeneity (0.4 < score $\leq 0.7$), and red dots indicate high heterogeneity (0.7 < score $\leq 1$).

After removing null values, the sample size in Figure 3 was determined by the number of corresponding in situ measurement sites with available seasonal heterogeneity scores during each unfrozen/frozen period. In Naqu, many grid cells had low heterogeneity scores, ranging from 0.12 to 0.40, with the number of low-heterogeneity grids accounting for 79% (42/53), 70% (38/54), 58% (32/55), 64% (35/55), and 53% (28/53) in the five temporal periods. The corresponding percentages of low-heterogeneity pixels in Maqu were 53% (8/15), 53% (8/15), 42% (6/14), 73% (8/11), and 50% (7/14), while for Ngari they were 50% (6/12), 0 (0/12), 17% (2/12), 0 (0/10), and 0 (0/12). Therefore, Figure 3 demonstrates that,

on average, Naqu contains less spatial heterogeneity than Maqu, and Maqu contains less than Ngari.

By excluding in situ measurement sites without data from April to September in 2016, the remaining grids with in situ measurement sites within the same network were sorted by their seasonal heterogeneity scores. Only the last few lowest-scoring grids in each network—i.e., eight from Naqu, five from Maqu, and five from Ngari—are listed in Table 2. Each grid is named after the corresponding in situ node name. Table 2 shows that, among the lowest-scoring in situ measurement sites, the score values still vary between in situ measurement sites and periods. However, some grids maintained relatively stable low spatiotemporal heterogeneity (e.g., P3, MS3494, and MS3518 in Naqu, NST03 in Maqu, and SQ06 in Ngari). In situ SM datasets from these in situ measurement sites were then selected for fine-RES SM validation.

**Table 2.** Results of the lowest-scoring spatiotemporal heterogeneity of the $0.01° × 0.01°$ grids with in situ measurement sites *.

| Network | Rank | 2014, Unfrozen | | 2014–2015, Frozen | | 2015, Unfrozen | | 2015–2016, Frozen | | 2016, Unfrozen | |
|---|---|---|---|---|---|---|---|---|---|---|---|
| | | Grid | $Score_{pixel}$ | Grid | $Score_{pixel}$ | Grid | $Score_{pixel}$ | Grid | $Score_{pixel}$ | Grid | $Score_{pixel}$ |
| Naqu | Lowest 8 sites | MS3518 | 0.1789 | P11 | 0.2822 | F4 | 0.2489 | MSBJ | 0.3143 | P11 | 0.3226 |
| | | MS3501 | 0.1774 | MS3488 | 0.2755 | BC05 | 0.2302 | F2 | 0.3141 | MS3523 | 0.3171 |
| | | MS3533 | 0.1694 | MS3494 | 0.2721 | MS3593 | 0.2114 | F1 | 0.3122 | C2 | 0.3090 |
| | | P11 | 0.1653 | C2 | 0.2683 | BC07 | 0.1735 | C2 | 0.3104 | MSBJ | 0.3054 |
| | | MS3494 | 0.1592 | MSBJ | 0.2681 | MS3494 | 0.1650 | P3 | 0.3063 | P3 | 0.2999 |
| | | BC07 | 0.1496 | P3 | 0.2410 | P3 | 0.1459 | MS3494 | 0.2901 | MS3518 | 0.2968 |
| | | MS3523 | 0.1311 | BC07 | 0.2329 | MS3518 | 0.1426 | BC07 | 0.2841 | MS3494 | 0.2950 |
| | | P3 | 0.1198 | MS3518 | 0.2132 | MS3523 | 0.1420 | MS3518 | 0.2797 | BC07 | 0.2744 |
| Maqu | Lowest 5 sites | NST-07 | 0.7924 | NST-07 | 0.7045 | NST-02 | 0.8563 | NST-08 | 0.6754 | NST-07 | 0.7929 |
| | | NST-01 | 0.7853 | NST-08 | 0.6260 | NST-07 | 0.7988 | NST-05 | 0.6345 | NST-01 | 0.4082 |
| | | NST-08 | 0.4246 | NST-01 | 0.4822 | NST-08 | 0.6553 | NST-06 | 0.4537 | NST-08 | 0.3974 |
| | | NST-03 | 0.1130 | NST-09 | 0.3337 | NST-09 | 0.4188 | NST-09 | 0.3189 | NST-09 | 0.2988 |
| | | NST-09 | 0.0759 | NST-03 | 0.2516 | NST-03 | 0.2694 | NST-03 | 0.2830 | NST-03 | 0.2111 |
| Ngari | Lowest 5 sites | SQ02 | 0.8211 | SQ02 | 0.8143 | SQ02 | 0.6524 | SQ10 | 0.7016 | SQ10 | 0.6939 |
| | | SQ10 | 0.6258 | SQ10 | 0.6383 | SQ10 | 0.6319 | SQ08 | 0.6359 | SQ01 | 0.6197 |
| | | SQ01 | 0.3195 | SQ01 | 0.4942 | SQ14 | 0.6093 | SQ01 | 0.5395 | SQ08 | 0.5293 |
| | | SQ06 | 0.3152 | SQ06 | 0.4480 | SQ01 | 0.5833 | SQ06 | 0.4911 | SQ06 | 0.4496 |
| | | SQ14 | 0.2773 | SQ14 | 0.4108 | SQ06 | 0.3936 | SQ14 | 0.4512 | SQ14 | 0.4454 |

* Red indicates high rank, while blue indicates low rank. The darker the color, the higher/lower its spatiotemporal heterogeneity value.

### 4.2. Direct Validation with In Situ Datasets

The temporal accuracy of the fine-RES SM datasets was assessed via comparisons with in situ SM measurements from the selected in situ measurement sites with low and stable heterogeneity in the three networks over the period of 1 April 2016 to 31 December 2016. Based on our above analysis of the spatial representativeness of the ground measurements, the direct validation of the fine-RES SM datasets was only implemented over the selected grids identified in Section 4.1.

Figure 4 shows the temporal evolutions of $0.01° × 0.01°$ fine-RES SM products over the selected grids, with precipitation plotted as an auxiliary time series. For the Naqu P3 grid cell, the ATA-OWCM and ER-OWCM time series accurately capture the in situ SM dynamic change, and are closer to the ground measurements than the corresponding GM-derived SM product in spring. RF-GM SM overestimates SM throughout the period, and NN-GM SM is too flat in the summer. For the Naqu MS3494 and MS3518 grid cells, GM-derived SM poorly follows the ground measurements—especially during pluvial summer conditions. In contrast, OWCM-based estimates are generally better. In the Maqu NST03 grid cell, for example, fine-RES SM is distinctly underestimated by GMs, while OWCM-derived SM accurately captures sharp temporal changes associated with observed precipitation.

Over the Ngari SQ06 grid cell, the fine-RES SMs derived from all eight methods follow the observed in situ SM trend. Note that OWCM-derived SM products perform better than GM-derived SM products—that is, they exhibit greater consistency with in situ observations, and relatively better dynamics. Overall, it can be observed that the OWCM-derived SM products are generally closer to the in situ SM, and are well matched with in situ SM dynamic changes during the wet season, while the GM-derived SM results show fewer temporal changes and more stable trends. These differences can likely be attributed to the improved fitting of OWCMs within the WT space.

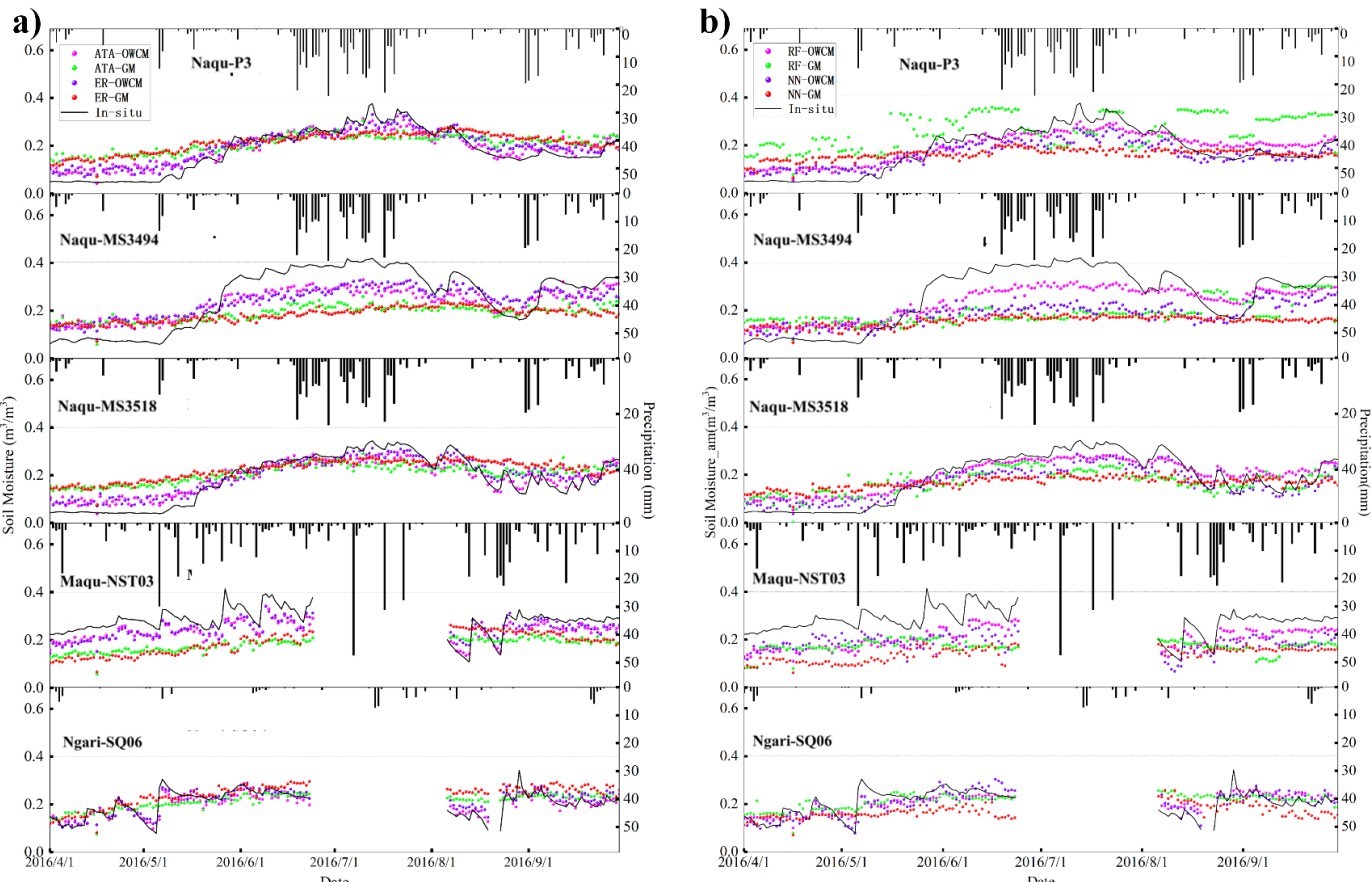

**Figure 4.** Time-series comparisons of daily GM-/OWCM-derived fine-RES SM with in situ SM measurements: (**a**) GM and OWCM results based on ATA and ER; (**b**) GM and OWCM results based on RF and NN.

Figure 4 also shows the mismatch between GM-/OWCM-derived SM and in situ SM. GM- and OWCM-derived SM products are generally underestimated in the wet environment (Maqu) or season (summer), and slightly overestimated in the relatively dry environment (Ngari) or seasons (spring and autumn). This likely relates to the differences in the soil layer observation depths of satellite (1–2 cm) and in situ measurements (~5 cm). These different observation depths can lead to (1) an underestimation of SM by satellite in the dry season, as the upper soil layers are drier than the deeper layers; and (2) an overestimation in the wet season (summer) or soon after a rain event, because the upper layers are wetter than the deeper layers [9,44,90]. By separating the high- and low-frequency feature information of SM and SM fitting factors, OWCMs may be able to obtain high-/low-frequency wavelet components of SM and SM fitting factors that can reflect SM information in the deeper layers, so as to obtain more consistent verification results with ground observations [44].

Table 3 shows the classical statistical validation metrics, including correlation coefficient (R), bias, and root-mean-square error (RMSE), as well as unbiased root-mean-square error (ubRMSE). Compared with GM-derived SM, the corresponding OWCM-derived SM shows a significant improvement via higher R and lower RMSE values against in situ SM measurements. Taking the three Naqu measurement sites as examples, the R and RMSE values between RF-OWCM SM and in situ SM range from 0.720 to 0.970, and from 0.039 $m^3/m^3$ to 0.070 $m^3/m^3$, respectively. By contrast, the R and RMSE values between RF-GM SM and in situ SM range from 0.203 to 0.437, and from 0.106 $m^3/m^3$ to 0.138 $m^3/m^3$, respectively. This indicates that the correlation between RF-GM SM and the in situ SM is weak, and the corresponding RMSE is generally above an acceptable accuracy threshold of 0.040 $m^3/m^3$ (as defined by the NASA SMAP mission). The same is true for other GM-derived SMs. In contrast, OWCM-derived SM estimates meet this threshold at most in situ measurement sites. Compared to in situ SM, the ubRMSE of OWCM-derived SM over the selected in situ measurement sites ranges between 0.0165 $m^3/m^3$ and 0.0895 $m^3/m^3$—better than the corresponding ubRMSE of GM-derived SM, which ranges between 0.0300 $m^3/m^3$ and 0.1209 $m^3/m^3$.

**Table 3.** Statistical metrics of GM-/OWCM-derived fine-RES SM datasets based on comparisons with in situ SM measurements. The superior metric within each pair of corresponding GM and OWCM products is indicated by bold type.

| | | Bias ($m^3/m^3$) | | R | | RMSE ($m^3/m^3$) | | ubRMSE ($m^3/m^3$) | |
|---|---|---|---|---|---|---|---|---|---|
| | | OWCM | GM | OWCM | GM | OWCM | GMs | OWCM | GMs |
| Naqu-P3 | ER | **0.0176** | 0.0331 | **0.9692** | 0.8225 | **0.0354** | 0.0695 | **0.0307** | 0.0613 |
| | ATA | **0.0108** | 0.0289 | **0.9546** | 0.8374 | **0.0370** | 0.0689 | **0.0354** | 0.0627 |
| | RF | **0.0106** | 0.0798 | **0.9519** | 0.4368 | **0.0385** | 0.1175 | **0.0371** | 0.0865 |
| | NN | **−0.0153** | −0.0174 | **0.9587** | 0.6723 | **0.0460** | 0.0792 | **0.0435** | 0.0775 |
| Naqu-MS3494 | ER | **−0.0187** | −0.0764 | **0.7122** | 0.7007 | **0.0747** | 0.1283 | **0.0725** | 0.1034 |
| | ATA | **−0.0255** | −0.0683 | **0.7412** | 0.4086 | **0.0730** | 0.1192 | **0.0686** | 0.0980 |
| | RF | **−0.0247** | −0.0664 | **0.7204** | 0.2031 | **0.0698** | 0.1376 | **0.0655** | 0.1209 |
| | NN | **−0.0759** | −0.1055 | **0.6908** | 0.3898 | **0.1172** | 0.1497 | **0.0895** | 0.1065 |
| Naqu-MS3518 | ER | **0.0122** | 0.0355 | **0.9633** | 0.8410 | **0.0378** | 0.0769 | **0.0359** | 0.0684 |
| | ATA | **0.0029** | 0.0204 | **0.9621** | 0.8591 | **0.0383** | 0.0724 | **0.0383** | 0.0697 |
| | RF | **0.0110** | 0.0431 | **0.9703** | 0.3298 | **0.0416** | 0.1059 | **0.0402** | 0.0970 |
| | NN | −0.0211 | **−0.0208** | **0.9645** | 0.8037 | **0.0452** | 0.0833 | **0.0401** | 0.0809 |
| Maqu-NST03 | ER | **−0.0345** | −0.0911 | **0.7445** | 0.3446 | **0.0494** | 0.1163 | **0.0355** | 0.0726 |
| | ATA | **−0.0424** | −0.0980 | **0.8922** | 0.1053 | **0.0496** | 0.1134 | **0.0258** | 0.0572 |
| | RF | **−0.0768** | −0.1067 | **0.7079** | 0.2487 | **0.0855** | 0.1217 | **0.0377** | 0.0587 |
| | NN | **−0.0934** | −0.1392 | **0.8038** | 0.1798 | **0.0987** | 0.1500 | **0.0321** | 0.0560 |
| Maqu-NST09 | ER | **0.0083** | 0.0346 | **0.8713** | 0.5545 | **0.0273** | 0.0583 | **0.0261** | 0.0470 |
| | ATA | −0.0084 | **0.0076** | **0.8890** | 0.5340 | **0.0260** | 0.0458 | **0.0247** | 0.0453 |
| | RF | **−0.0010** | 0.0068 | **0.7627** | 0.4211 | **0.0341** | 0.0493 | **0.0342** | 0.0490 |
| | NN | **−0.0036** | −0.0352 | **0.7937** | 0.1686 | **0.0335** | 0.0665 | **0.0335** | 0.0566 |
| Ngari-SQ06 | ER | **−0.0340** | −0.0853 | **0.8122** | 0.0242 | **0.0398** | 0.0922 | **0.0208** | 0.0349 |
| | ATA | **−0.0418** | −0.0894 | **0.7748** | 0.1711 | **0.0472** | 0.0959 | **0.0219** | 0.0348 |
| | RF | **−0.0541** | −0.1304 | **0.7879** | 0.1463 | **0.0584** | 0.1354 | **0.0220** | 0.0366 |
| | NN | **−0.0425** | −0.0958 | **0.7717** | 0.0915 | **0.0481** | 0.1025 | **0.0227** | 0.0365 |
| Ngari-SQ14 | ER | **0.0041** | 0.0565 | **0.6734** | 0.1124 | **0.0232** | 0.0644 | **0.0229** | 0.0309 |
| | ATA | 0.0074 | **0.0358** | **0.8455** | 0.2705 | **0.0180** | 0.0466 | **0.0165** | 0.0300 |
| | RF | **−0.0025** | 0.0491 | **0.7261** | 0.0054 | **0.0217** | 0.0585 | **0.0216** | 0.0319 |
| | NN | **−0.0141** | 0.0455 | **0.7850** | 0.0470 | **0.0237** | 0.0578 | **0.0191** | 0.0358 |

This significant improvement has two likely sources, associated with both the application of OWCMs—which implement the fitting procedures in the WT space—as well as the selection of grid cells with low spatial heterogeneity: First, the WT applied in the

OWCMs separates high- and low-frequency SM features in a way that better captures the relationship between SM and various land-surface attributes. Second, in situ measurement sites with lower spatial heterogeneity demonstrate better representativeness of the overall grid cell and, therefore, minimize error associated with point-to-grid-cell upscaling.

## 5. Discussion

### 5.1. Discussion of the Consistency between the OWCM-/GM-Derived SM and RFSM

#### 5.1.1. Spatial Consistency

Typically, renormalization is included in the downscaling procedures so that spatial resampling of fine-RES SM back to the original coarse-resolution grid will exactly match the original coarse-scale products being downscaled (e.g., RFSM in this case). However, it is worth noting that the eight downscaled fine-RES SM products obtained by GMs and OWCMs were not renormalized in this way, so that we could better understand the relative performance of the GMs and OWCMs.

Within the QTP, Figure 5 shows images of monthly averaged SM estimates from corresponding GM- and OWCM-derived fine-RES SM products, along with the original RFSM. All eight fine-RES SM products generally match the temporal dynamic changes of the original RFSM. However, SM estimates derived from the ATA-GM and ATA-OWCM approaches are consistently better matched to the spatial patterns of RFSM; RF-GM and RF-OWCM SM also have good performance, especially in the middle, south, and west areas in summer. Considering the residual function in the ATA approach, it is reasonable to infer that ATA has the best ability to reflect the spatial pattern of the original RFSM product accurately. For both RF-GM and RF-OWCM, despite the machine learning model often being characterized by a sliding and smoothing effect, a spurious mosaic phenomenon (i.e., the reflection of the underlying $0.25° \times 0.25°$ grid in fine-RES SM results) is still seen in certain areas. However, on the whole, the GM and OWCM of ER and NN have a stronger smoothing effect than those of RF, and underestimate the spatial variability present in the original RFSM product.

It can be observed that the most evident spatial characteristic of the original RFSM is the increasing trend of SM from west to east over the QTP. Higher SM typically occurs in the east, south, and southeast, while lower SM appears in the central and western regions. The dynamic range of RFSM is better reflected in the OWCM-derived SM products than in the GM-derived SM—especially for ER- and NN-based SM. Overall, the OWCM-derived SM datasets represent a significant improvement with respect to the corresponding GM-derived SM datasets, and the GM algorithms are deficient in preserving the dynamic range of coarse-resolution SM information [24].

Moreover, the R and ubRMSE between the aggregated OWCM-/GM-derived SM datasets and the original RFSM products were calculated spatially, as shown in Figure 6. This echoes the comparison results above. ATA-OWCM SM behaves nearly the same as ATA-GM SM, with a similar spatial distribution of R and ubRMSE. Moreover, the ATA-OWCM and ATA-GM SM products preserve the spatial characteristics of the original RFSM over most areas, with clearly higher R and lower ubRMSE compared with the other three GM- and OWCM-derived SM products. RF-OWCM SM shows overall higher R and lower ubRMSE than RF-GM SM. ER-OWCM SM also obtains overall better R values—especially over the west and middle of the QTP, where the ER-GM SM product does not perform well. Likewise, the ubRMSE of ER-OWCM SM shows a better spatial distribution than that of ER-GM SM compared with the original RFSM products. It can be observed that the NN-GM and NN-OWCM SM datasets have the worst performance among the four GM- and OWCM-derived SM datasets, with NN-OWCM still outperforming NN-GM.

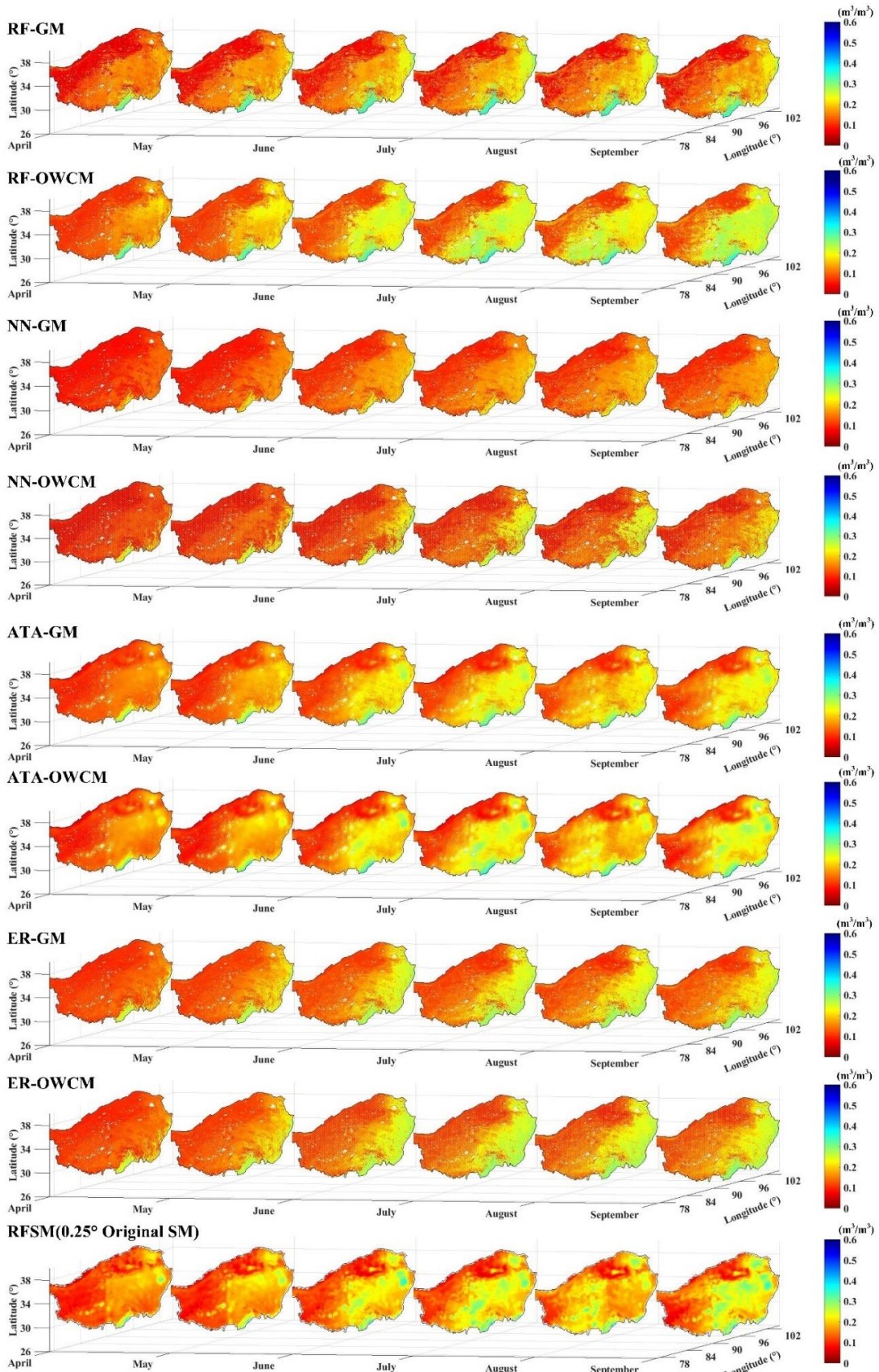

**Figure 5.** The spatial distribution of the monthly averaged SM products over the QTP during the 2016 unfrozen season, calculated from the corresponding GM-/OWCM-derived fine-RES SM, along with the original RFSM.

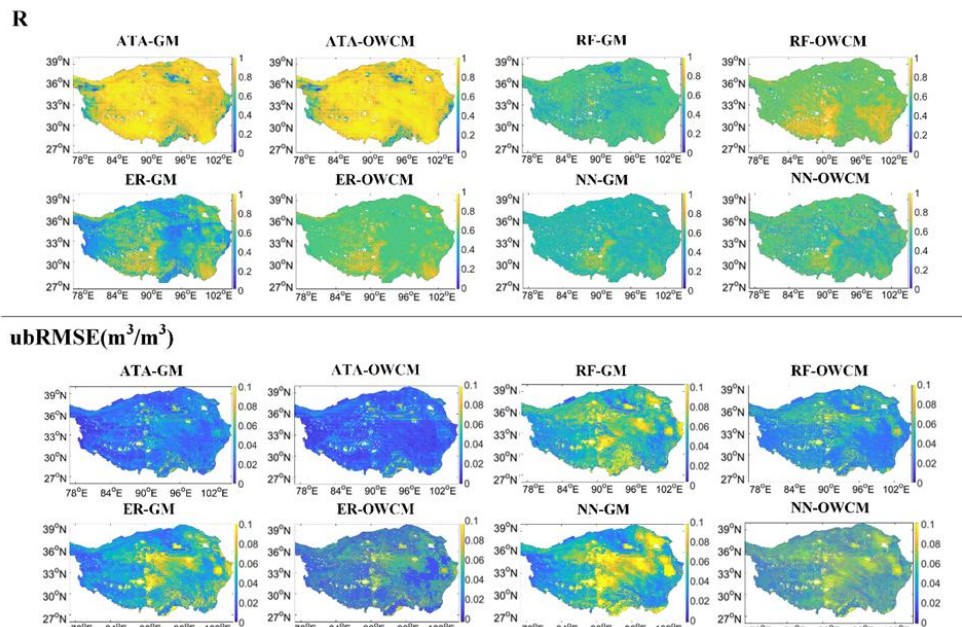

**Figure 6.** Spatial distributions of the statistical metrics R and ubRMSE between the aggregated OWCM-/GM-derived SM and the original RFSM.

As a summary of the information presented in Figure 6, Figure 7 shows the spatial frequency distributions of the statistical metrics R, ubRMSE, and bias between the aggregated OWCM-/GM-derived SM and the original RFSM over the QTP. In general, the distribution of OWCM-derived SM metrics is notably superior to that of the corresponding GM-derived SM products, and the ATA-OWCM SM products perform the best overall.

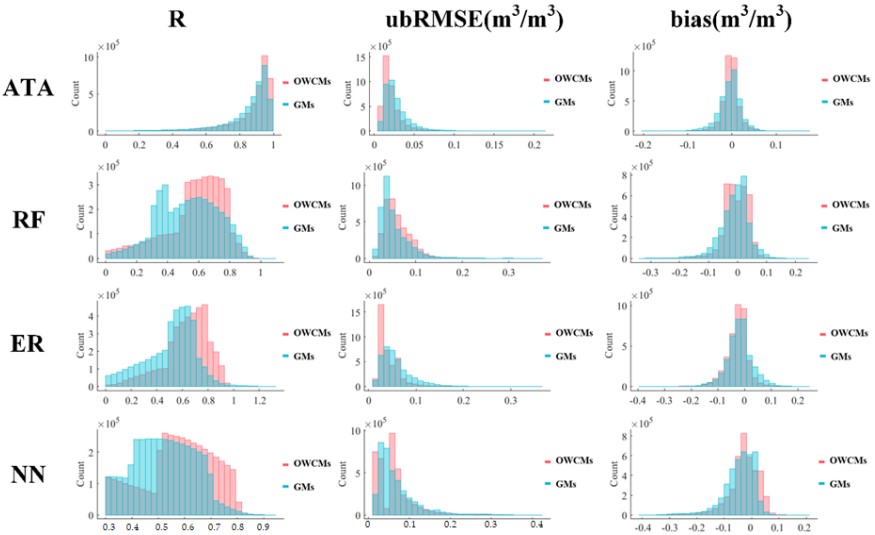

**Figure 7.** Frequency distributions of the statistical metrics R (1st column), ubRMSE (2nd column), and bias (3rd column) between the ATA (1st row)/RF (2nd row)/ER (3rd row)/NN (4th row)-based OWCM/GM SM and the original RFSM over the QTP.

### 5.1.2. Temporal Consistency

EDA was utilized to test the time-series correlation R between the OWCM-/GM-derived SM and the original RFSM in different time windows over the Naqu network. The results are shown in Figure 8. Overall, in Naqu, the OWCMs performed significantly better than GMs, and had a certain degree of optimization to the original method. Over the entire period and the local periods, the correlation between the GM-derived SM and

the original RFSM was improved to some extent by the OWCM when the correlation between GM-derived SM and the original RFSM was poor (or even negative). At the same time, it can be observed that the periods of poor correlation with the original RFSM for all OWCMs and GMs were generally confined to the central days (*x*-axis) from 90 to 160—corresponding to the period from June to August in the summer. According to Section 4.2, OWCM-/GM-derived SM was generally lower than the SM measured in situ during this period. However, it is worth noting that such underestimation is more significant for GMs than for OWCMs, indicating the improvement in the underestimation by replacing GMs with the corresponding OWCMs.

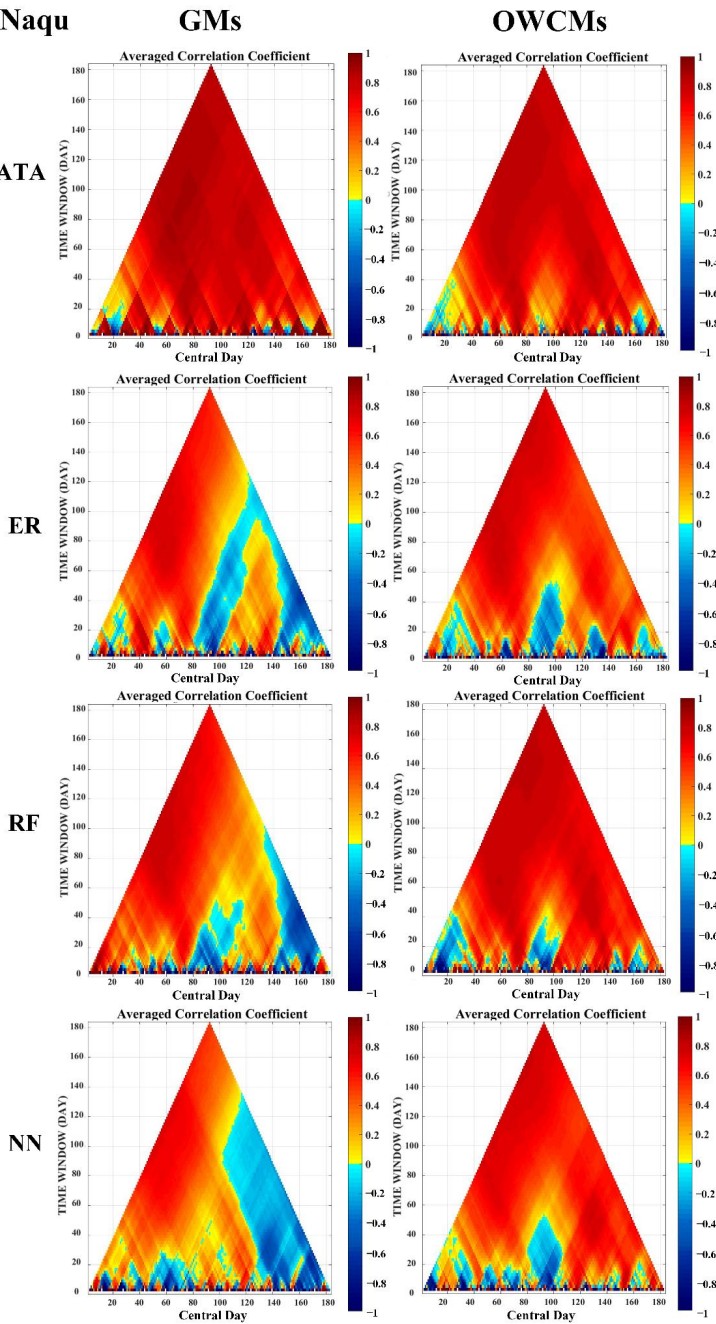

**Figure 8.** EDA-based average correlation coefficient R plots between GM-/OWCM-derived SM and the original RFSM in Naqu.

### 5.2. Discussion of the Impacts of Spatial Heterogeneity on the Fitting Accuracy

According to Section 3.3, the impacts of spatial heterogeneity on the fitting accuracy of OWCMs and GMs are further analyzed and discussed. Based on GAM, the partial deviance explained by each factor (e.g., $CV_{LAI}$, $CV_{LST}$, $CV_{TVDI}$ and $CV_{SEE}$) for the fitting accuracy of each OWCM/GM was determined. Since the spline change trends for the fitting accuracies of the eight GMs and OWCMs are similar, only the top four explained deviances are plotted in Figure 9 for a clear presentation. The gray shaded area indicates 90% confidence intervals. Unless otherwise specified, all discussion below is based on these intervals.

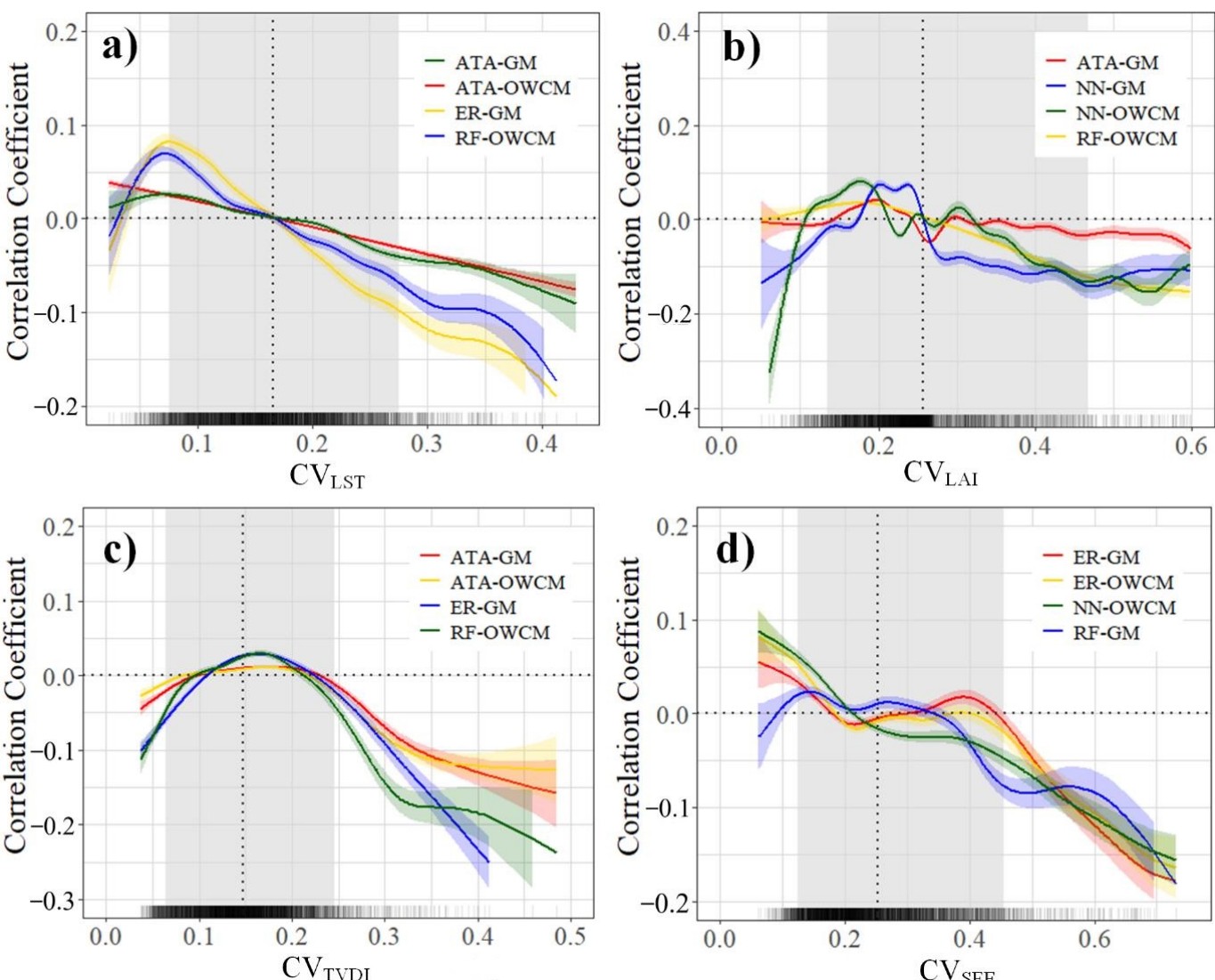

**Figure 9.** GAM partial effect splines presenting the dynamic changes in the fitting accuracy R with (**a**) $CV_{LST}$, (**b**) $CV_{LAI}$, (**c**) $CV_{TVDI}$, and (**d**) $CV_{SEE}$. In (**a–d**), the dotted lines indicate the zero value (*y*-axis) and the means of explanatory variables (*x*-axis), while the shaded areas are the 90% confidence intervals. Moreover, the small vertical lines at the bottoms of the figures represent data points.

Overall, spatial heterogeneity shows a substantial impact on the fitting accuracy. The contribution to the correlation coefficient R (between aggregated GM-/OWCM-derived SM and RFSM) decreases from positive (or around zero) to negative as the spatial heterogeneity increases—especially under high-CV conditions. Specifically, when $CV_{LST}$ increases, the explained partial deviance decreases almost monotonously (Figure 9a). This indicates that the LST heterogeneity negatively impacts the fitting accuracy, i.e., a higher $CV_{LST}$ corresponds to lower fitting accuracy. LST has been regarded as a strong indicator of SM

in previous studies due to the controlling effect of SM on surface energy exchange and partition [37,91]. The pixel spatial heterogeneity brings uncertainty to the SM fitting process because the pixel values of LST at different scales might not represent one another well [85]. Figure 9b shows that the explained partial deviances fluctuate around zero for small $CV_{LAI}$ values, but decrease slightly when $CV_{LAI}$ increases. This implies that low vegetation heterogeneity has no significant impact on the fitting accuracy, but still has a solid negative impact during conditions of high vegetation heterogeneity. This could be explained by the more complex couplings and feedback between SM and vegetation conditions (i.e., LAI) when LAI is distributed heterogeneously within the pixel [92]. Figure 9c shows four upside-down 'U' spline shapes for $CV_{TVDI}$, with the explained partial deviance shifting from negative to positive, and then back to negative. This indicates that the fitting accuracies are less affected by moderate $CV_{TVDI}$ levels, but substantially negatively affected by low- and high-$CV_{TVDI}$ conditions. In Figure 9d, the explained partial splines for $CV_{SEE}$ show a clear downward trend similar to those for $CV_{LST}$, drawing the same conclusions as discussed above for $CV_{LST}$.

## 6. Conclusions

Aiming to improve the fitting accuracies of the general models used in SM analyses (e.g., downscaling, interpolation, and forecasting), the WT technique was introduced here to implement the required fitting procedures in the WT space rather than the regular space. As a test, four general regression models—i.e., ATA, ER, RF and NN—were selected and applied to the specific SM problem of spatial downscaling. By coupling (OWCMs) and uncoupling (GMs) the wavelet transformation, empirical relationships between the $0.25° \times 0.25°$ RFSM and $0.01° \times 0.01°$ GLASS FVC, LAI, albedo, and TRIMS LST were constructed. As a result, eight downscaled fine-RES SM datasets were produced using the trained GMs and OWCMs.

The performances of the GMs and OWCMs were demonstrated by validating and analyzing the fine-RES SM datasets from two aspects: One was based on direct comparisons with in situ SM measurements. The other was based on indirect intercomparisons of the spatial and temporal consistencies between the aggregated OWCM-/GM-derived SM and the original RFSM dataset. Overall, the OWCM-derived SM was generally closer to the in situ SM and better matched with the in situ SM dynamic change during the unfrozen season compared to the corresponding GM-derived SM products, showing fewer time changes and more stable trends. Moreover, the OWCM-derived SM represents a significant improvement over the corresponding GM-derived SM regarding the RFSM in terms of spatial distribution and temporal variation. As discussed in Section 4.2, these improvements can possibly be attributed to two main factors, i.e., the application of OWCMs, which implement the fitting procedures in the WT space, and the selection of grid cells with low spatial heterogeneity. Since WT separates the high- and low-frequency feature information, implementing the fitting procedure between the same WT components of SM and the same attributes can better capture the relationships between them. In addition, for both GM- and OWCM-derived SM products, an in situ measurement site with lower spatial heterogeneity has better representativeness of the overall grid cell and, therefore, minimizes errors due to the point versus grid-cell upscaling differences, consequently improving our statistical indices, i.e., generally higher R values and lower RMSEs and ubRMSEs.

Since the direct comparisons with in situ SM measurements were implemented only over grids with low spatial heterogeneity, the impacts of spatial heterogeneity (indicated by $CV_{LST}$, $CV_{LAI}$, $CV_{TVDI}$, and $CV_{SEE}$) on the fitting accuracy of OWCMs and GMs—i.e., the correlation coefficient R between aggregated GM-/OWCM-derived SM and RFSM—were further analyzed and discussed based on a GAM. The results show that, despite the selection of only spatially representative sites for validation, spatial heterogeneity still substantially impacts the fitting accuracy of both GMs and OWCMs. Generally, the contribution to the correlation coefficient R decreases from positive (or around zero) to negative as the spatial heterogeneity increases. It is worth noting that the method of characterizing the spatial heterogeneity to identify the in situ nodes with maximal spatial

representativeness, as described in Section 3.2.1, is arguably subjective due to its selection of the empirical coefficients adopted in Equations (4)–(6). However, the use of alternative coefficients does not significantly influence the spatial heterogeneity ranking over the grids with in situ measurement sites. OWCMs significantly improve the fitting effect of soil moisture on the QTP. In the future, it will be necessary to further verify this optimization effect in other regions and in situ networks, to determine its applicability in different related SM analysis topics (e.g., gap-filling, forecasting, and downscaling) that require SM fitting models.

**Author Contributions:** L.C. and Z.H. designed the work. Z.H. collected and processed the in-situ data, realized the idea, analyzed the results, and drafted the manuscript. L.C. and W.T.C. supervised and improved the research scheme, refine the ideas, reviewed and corrected the manuscript. S.L., Z.Z. and J.Z. contributed the ideas and methods in the discussion section. J.L., S.Y., Y.Q. and Z.L. contributed by providing additional analysis, and finalizing the paper. L.C. acquired the funding. All authors have read and agreed to the published version of the manuscript.

**Funding:** The work was supported by the Strategic Priority Research Program of the Chinese Academy of Sciences (XDA20100101), the National Natural Science Foundation of China (42171319, 41871241), and the State Key Laboratory of Earth Surface Processes and Resource Ecology (2021-ZD-04). The presentation, findings, and conclusions in this publication are those of the authors, and should not be construed to represent any official USDA or U.S. Government determination or policy. The USDA ARS is an equal opportunities employer.

**Conflicts of Interest:** The authors declare no conflict of interest.

## Appendix A. Multifactorial Statistic Factor Selection (MFS) Process

Anterior to operating the GMs and OWCMs in the SM fitting procedure, we utilized the MFS process to determine the appropriate fitting factors for the GMs and OWCMs. MFS can be regarded as an evaluation of the effectiveness of fitting factors.

MFS evaluates four fitting factors, along with their combinations (i.e., the product of every two factors, the square and cube of every factor), within four typical land cover types over the QTP, i.e., grasslands, croplands, barren or sparsely vegetated, and mixed forest, according to the IGBP classification scheme (MODIS MCD12Q1 product). For each pixel of a coarse WT component ($0.5° \times 0.5°$), MFS is directly applied in aggregated pure pixels of the four typical land cover types, where there is only one typical land cover type of the four. For other mixed pixels, the land cover type is regarded as the type of which the IGBP original pixels ($0.01° \times 0.01°$) account for the most significant proportion within it. For each pixel of a fine-resolution WT component ($0.02° \times 0.02°$), its land cover type is judged by the type accounting for the largest proportion within the $3 \times 3$ neighborhood pixels ($0.06° \times 0.06°$).

The MFS process involves two steps: (1) To identify the potentially relevant factors by comparing the consistency between the frequency distributions of each fitting factor and each WT-derived SM component in the probability density figure. (2) To determine the final fitting factors significantly correlated with each WT-derived SM component by testing partial correlation coefficients at 95% confidence intervals when fixing other fitting factors. Taking the most widely distributed land cover type of grassland as an example, the frequency distribution plots in the probability density of the four WT components (LL, HL, LH, and HH) are shown in Figure A1. For brevity, only the final selected factors are shown. Table A1 lists the corresponding fitting factors selected for the four WT components and their corresponding partial correlation coefficients over the four land cover types, i.e., grassland, cropland, barren or sparsely vegetated, and mixed forest. For GMs, the final selected factors are the same as the LL component, since LL is watched as the approximation image of the original.

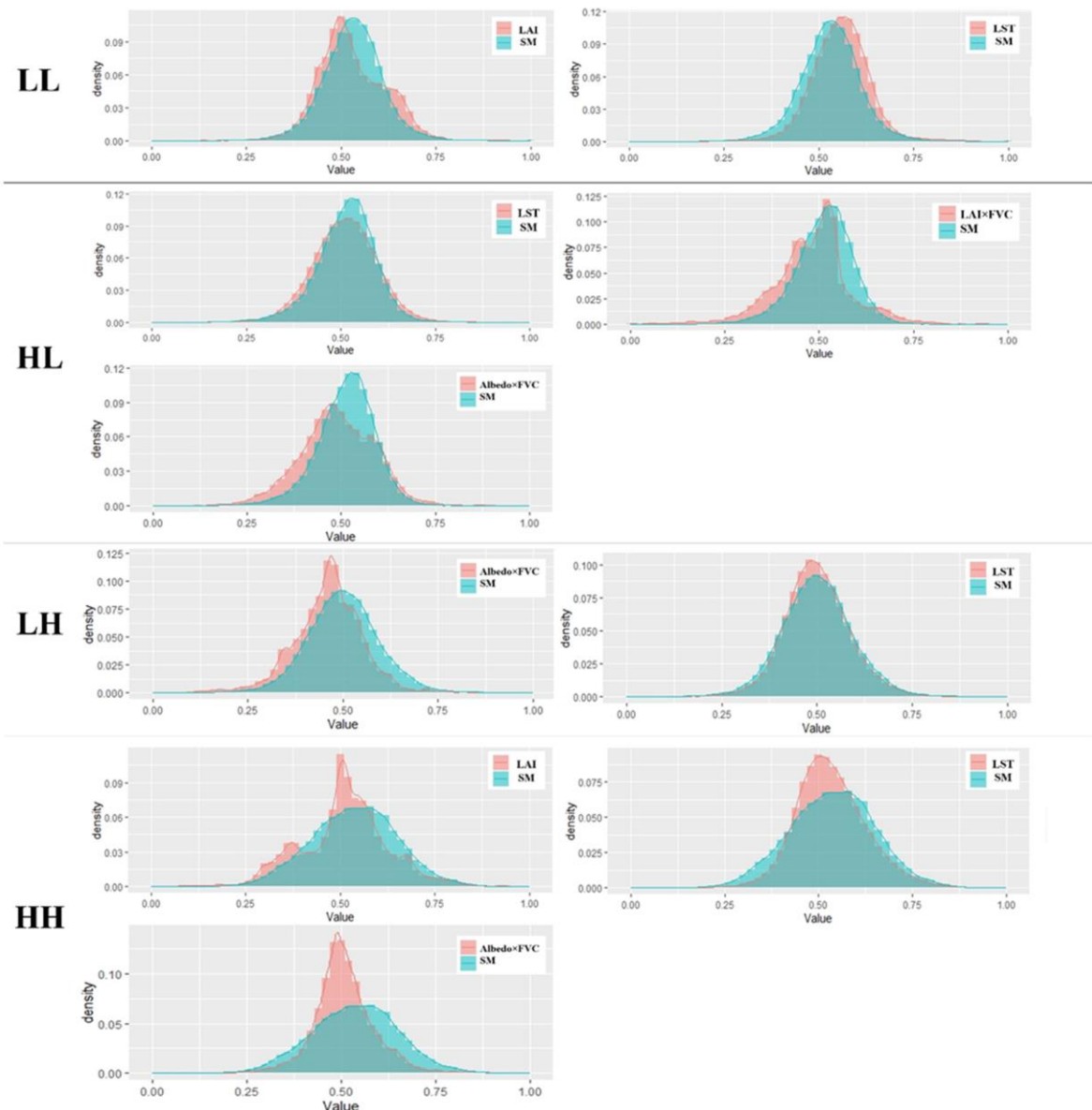

**Figure A1.** Frequency distributions in the probability density of the final selected fitting factors, with relatively good consistency with the frequency distribution of the original RFSM in the four WT components (LL, LH, HL, HH) over grassland.

**Table A1.** Summary of the final selected factors by MFS for the four land cover types over the QTP.

| Land Cover Types | WT Component | Selected Factors | Partial Correlation Coefficient * |
|---|---|---|---|
| Grasslands | LL | LAI, LST | 0.6673, 0.7247 |
| | HH | LAI, LST, albedo × FVC | 0.5487, 0.6924, 0.5765 |
| | HL | LST, albedo × FVC, LAI × FVC | 0.4982, 0.4792, 0.6593 |
| | LH | Albedo × FVC, LST | 0.7873, 0.6169 |
| Croplands | LL | FVC, LST | 0.8147, 0.7571 |
| | LH | Albedo × LAI, LAI × FVC, albedo × FVC | 0.6734, 0.5186, 0.4795 |
| | HL | FVC × LST | 0.6575 |
| | HH | Albedo × LAI, albedo × LST | 0.5917, 0.4452 |

**Table A1.** *Cont.*

| Land Cover Types | WT Component | Selected Factors | Partial Correlation Coefficient * |
|---|---|---|---|
| Barren or sparsely vegetated | LL | Albedo, LST | 0.7129, 0.6122 |
| | LH | Albedo × LAI | 0.7642 |
| | HL | Albedo × LAI | 0.6436 |
| | HH | Albedo × LAI, FVC × LST, LST × LAI | 0.5749, 0.6352, 0.4557 |
| Mixed forest | LL | FVC, LST | 0.6485, 0.5847 |
| | LH | Albedo × FVC | 0.5373 |
| | HL | LST × LAI | 0.5467 |
| | HH | Albedo × FVC, LAI × FVC, albedo × LST | 0.6278, 0.6617, 0.7423 |

* Partial correlation coefficient at 95% confidence interval.

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
