# Peer review of "Applying a Wavelet Transform Technique to Optimize General Fitting Models for SM Analysis: A Case Study in Downscaling over the Qinghai–Tibet Plateau"

_remotesensing, doi:10.3390/rs14133063_

Round 1

Reviewer 1 Report

The article is interesting and could make an important contribution to the field, but requires a strong development of the discussions and an international exposure required for publication in an international journal. Detailed comments are provided for each section of manuscript.
Figure 1 shows the inability of authors to write up research. This is an article for an international journal, and not a report for the national authorities. The authors should present a map showing the location of the study area in an international context, making visible the neighboring countries with their names, so that a Brazilian researcher could understand it too.
The most important section of a research article, the Discussions, meant to emphasize the importance of research, justifying its publication, is insufficiently developed, and this shortcoming is masked by integrating it with the results. The discussions must include (A) the significance of results - what do they say, in scientific terms; (B) the inner validation of results, against the study goals or hypotheses; (C) the external validation of results, against those of similar studies from other countries, identified in the literature; (D) the importance of the results, meaning their contribution (conceptual or methodological) to the theoretical advancement of the field; (E) a summary of the study limitations and directions for overcoming them in the future research. Out of all these, only the significance of results is presented to some extent; the other parts need to be developed too.

Author Response

Response to Reviewer 1 Comments

Thanks for your comments for the article improvement. I have revised and answered according to the comments.

Minor comments:

Point 1: Figure 1 shows the inability of authors to write up research. This is an article for an international journal, and not a report for the national authorities. The authors should present a map showing the location of the study area in an international context, making visible the neighboring countries with their names.

Response 1: Thanks for the comments. I have added the visible information of the neighboring countries with their names in Figure 1.

Point 2: The most important section of a research article, the Discussions, meant to emphasize the importance of research, justifying its publication, is insufficiently developed, and this shortcoming is masked by integrating it with the results. The discussions must include (A) the significance of results - what do they say, in scientific terms; (B) the inner validation of results, against the study goals or hypotheses; (C) the external validation of results, against those of similar studies from other countries, identified in the literature; (D) the importance of the results, meaning their contribution (conceptual or methodological) to the theoretical advancement of the field; (E) a summary of the study limitations and directions for overcoming them in the future research. Out of all these, only the significance of results is presented to some extent; the other parts need to be developed too.

Response 2: Thanks for the comments. I have modified the content of discussion section and conclusion section to emphasize the importance of research, add the inner/external validation of results and clarify the study limitations and directions in the last paragraph of the conclusion.

Reviewer 2 Report

Dear Authors,

please find my remarks on your manuscript. Congratulations on a well-done work.

Kind regards

Reviewer

Author Response

Response to Reviewer 2 Comments

Thanks for your comments for the article improvement. I have revised and answered according to the comments.

Minor comments:

Point 1: Lines 113-114: please provide some characteristics of this so-called‘unique environment’

Response 1: Thanks for your reminder. I have modified the content and provided some characteristics to describe the QTP environment in Lines 117-122.

Point 2: Figure 1: quality of the graph is very low. Legend for elevation should contain at least one intermediate value - around 4000 or so. Information on the data source for land use is missing. I would like to see some comments on the spatial distribution of in situ network on different sites.

Response 2: Thanks for your comments. I have corrected Figure 1 for better quality and added the intermediate value in legend for elevation. The information on the data source for land use is available in Section 2.4 Auxiliary Data (Lines 197-198). The site distribution and specific locations of the three ground observation networks are clearly shown in Figure 1 and Lines 125-132.

Point 3: Figures 4, 5, 8: please provide graphs of better quality.

Response 3: Provided.

Point 4: please add some comments on the possible application of your approach in other parts of the World (concerning the specification on the investigated study site).

Response 4: Thanks for your comments. I have added the corresponding content in the last paragraph of the conclusion section (Lines 625-629).